# The Role of Proteomics and Phosphoproteomics in the Discovery of Therapeutic Targets and Biomarkers in Acquired EGFR-TKI-Resistant Non-Small Cell Lung Cancer

**DOI:** 10.3390/ijms24054827

**Published:** 2023-03-02

**Authors:** Sutpirat Moonmuang, Apichat Tantraworasin, Santhasiri Orrapin, Sasimol Udomruk, Busyamas Chewaskulyong, Dumnoensun Pruksakorn, Parunya Chaiyawat

**Affiliations:** 1Center of Multidisciplinary Technology for Advanced Medicine (CMUTEAM), Faculty of Medicine, Chiang Mai University, Chiang Mai 50200, Thailand; 2Clinical Epidemiology and Clinical Statistics Center, Faculty of Medicine, Chiang Mai University, Chiang Mai 50200, Thailand; 3Clinical Surgical Research Center, Department of Surgery, Faculty of Medicine, Chiang Mai University, Chiang Mai 50200, Thailand; 4Musculoskeletal Science and Translational Research Center, Department of Orthopedics, Faculty of Medicine, Chiang Mai University, Chiang Mai 50200, Thailand; 5Division of Oncology, Department of Internal Medicine, Faculty of Medicine, Chiang Mai University, Chiang Mai 50200, Thailand

**Keywords:** lung cancer, non-small cell lung cancer, EGFR-TKI resistance, proteomics, phosphoproteomics

## Abstract

The discovery of potent EGFR-tyrosine kinase inhibitors (EGFR-TKIs) has revolutionized the treatment of EGFR-mutated lung cancer. Despite the fact that EGFR-TKIs have yielded several significant benefits for lung cancer patients, the emergence of resistance to EGFR-TKIs has been a substantial impediment to improving treatment outcomes. Understanding the molecular mechanisms underlying resistance is crucial for the development of new treatments and biomarkers for disease progression. Together with the advancement in proteome and phosphoproteome analysis, a diverse set of key signaling pathways have been successfully identified that provide insight for the discovery of possible therapeutically targeted proteins. In this review, we highlight the proteome and phosphoproteomic analyses of non-small cell lung cancer (NSCLC) as well as the proteome analysis of biofluid specimens that associate with acquired resistance in response to different generations of EGFR-TKI. Furthermore, we present an overview of the targeted proteins and potential drugs that have been tested in clinical studies and discuss the challenges of implementing this discovery in future NSCLC treatment.

## 1. Introduction

Lung cancer is the most common type of cancer and the main cause of cancer deaths globally, with non-small cell lung cancer (NSCLC) accounting for over 80% of all cases [1]. Although surgery is a very effective treatment option for treating early-stage NSCLC, most cases are diagnosed after the cancer has spread and surgical resection is no longer feasible, resulting in an unsatisfied overall 5-year relative survival rate of 26% and only 8% in metastasis [2]. Even though platinum-based chemotherapy (PBC) is a standard treatment for patients with advanced NSCLC, the outcomes are dismal, with an objective response rate (ORR) of around 30% and a median progression-free survival (PFS) of about 5–6 months [3,4,5].

The identification of oncogenic driver mutations in the epidermal growth factor receptor (EGFR) gene was a breakthrough in NSCLC diagnosis and treatment. These activating mutations, which occur in up to 50% of NSCLC patients, result in ligand-independent downstream signaling of EGFR, promoting increased malignant cell survival, proliferation, invasion, and metastasis [6]. Over the past decade, tyrosine kinase inhibitors (TKIs) have been recommended as a treatment for several types of cancers [7]. Among them, the inhibitor targeting EGFR tyrosine kinase (EGFR-TKI), which inhibits EGFR signaling overactivation, has demonstrated remarkable efficacy in NSCLC patients with EGFR-activating mutations. Although EGFR-TKIs have a satisfying therapeutic response that shifts NSCLC treatment strategy to a targeted strategy, most patients will develop the progressive disease within one year of treatment due to drug resistance [8].

Inherent and acquired resistance in EGFR-mutated lung adenocarcinomas constitutes a significant obstacle to improving lung cancer treatment outcomes [9]. The initial inefficacy of EGFR-TKIs is typically referred to as “intrinsic resistance”. Several studies of non-response to EGFR-TKIs have been reported in the context of non-classical sensitizing EGFR mutations and rarely in classical EGFR mutations, despite the fact that the mechanisms of intrinsic resistance are not fully investigated [9]. In-frame insertions of base pairs in exon 20 of the EGFR gene are the most common intrinsic resistance mechanisms to EGFR TKIs, accounting for 4–10% of all EGFR mutations observed in NSCLC [10]. Patients with EGFR exon 20 insertion had very poor response rates to erlotinib, gefitinib, and afatinib therapy, ranging from 3 to 8% [11]. Additionally, existing molecular or genetic changes that may possibly impair the sensitivity to EGFR-TKI therapy might result in intrinsic resistance [9]. The deletion polymorphisms or low levels of messenger RNA (mRNA) of the proapoptotic Bcl-2 family member, BIM, enabled the tumor to resist the apoptosis effects of EGFR-TKI [12].

The acquired EGFR T790M mutation in exon 20 is the most prominent alteration related to the emergence of resistance to the first and second generations of EGFR-TKI. Although Osimertinib, the third generation of EGFR-TKI, was employed to address lung cancer with the T790M mutation and has shown excellent effectiveness in this setting, the acquired resistance to the third generation of EGFR-TKI, which involves the cysteine residue at codon 797, has been observed. The activation of alternate pathways or downstream targets of EGFR signaling and histological transformation are additional acquired resistance mechanisms [13].

With the advancement of mass spectrometry (MS)-based protein analysis technology, large-scale protein analysis has become increasingly popular. Especially in cancer research, proteomic analysis of cancers is critical for gaining a comprehensive understanding of dynamic molecular aberrations, including protein phosphorylation, protein–protein interactions, protein structure, and protein function [14,15]. Discovery proteomics enables the detection of protein dynamics in biological states and pathological situations as well as the large-scale identification of proteins [16]. Furthermore, with advancements in technology for sample preparation and data processing, as well as increases in the sensitivity and resolution of MS instrumentation, such an approach has become a key technology for illustrating proteins related to cancer drug resistance [17]. These allow the discovery of novel therapies as well as potential biomarkers for predicting patient prognosis, stratifying high-risk patients, and responding to specific medicines.

Here, we emphasize the proteome and phosphoproteomic research of EGFR-TKI-resistant NSCLC cells, which gives significant details on the acquired resistance mechanisms for each EGFR-TKI generation. The mechanisms of EGFR-TKI resistance and prospective therapeutic approaches have been comprehensively described, including alterations in key signaling pathways as well as the metabolome and lipidome profiles of resistant cells [18,19,20,21]. In addition to the well-known acquired EGFR-TKI resistance, the proteome analysis sheds light on important resistant mechanisms, such as the posttranslational modifications of EFGR and antigen-presenting pathways. We also discuss the proteome analysis of biofluid samples, which have clinical potential as biomarkers for patient stratification and prognostic indication.

## 2. EGFR-TKIs

EGFR, also known as HER1, belongs to the ErbB family of receptor tyrosine kinases (RTK) [22], consists of an extracellular ligand binding domain for the EGF family, a single α-helical transmembrane domain, an intracellular tyrosine kinase domain, and a carboxy-terminal region that contains autophosphorylation sites. Upon ligand interaction, the dimerization of EFFR enhances its intracellular protein tyrosine kinase activity, resulting in autophosphorylation thereby activating signal cascades including RAS/RAF/MEK/ERK, PI3K/AKT/mTOR, and STAT pathways [23]. Exon 19-microdeletions (exon 19dels) or deletion-insertions (exon 19 delins) or the p.L858R (L858R) point mutation in exon 21 of EGFR, which accounts for roughly 90% of all EGFR mutations in NSCLC [6], were the most commonly associated with classic EGFR activating mutations. However, these mutations cause constitutive activation of ligand-independent downstream signaling of EGFR [24], causing enhanced malignant cell survival, proliferation, invasion, and metastasis (6). Thus, the application of TKIs targeting EGFR mutations has accelerated the evolution of NSCLC therapy. To date, EGFR-TKIs have been extensively explored and played critical roles in the treatment of EGFR-mutant NSCLC patients, as summarized in (Table 1).

### 2.1. First- and Second-Generation EGFR-TKIs

The conformational change that destabilizes the dormant form of the EGFR induced by the classical EGFR mutations results in constitutive activation of downstream signaling pathways [24]. The first-generation EGFR-TKIs target this conformational alteration, leading to the inhibition of EGFR signaling. The findings of randomized clinical trials in advanced NSCLC patients with classical EGFR mutations showed the outperforming of first-generation reversible EGFR-TKIs over platinum-based doublet chemotherapy (PBC) and a much larger increase in second-generation irreversible EGFR-TKIs. Regardless of the promising activity of first- and second-generation EGFR-TKIs, the acquired resistance due to EGFR T790M mutation hampered the efficacy of treatment by interfering with the binding of first- and second-generation TKIs to the ATP-binding site and has been identified approximately 50% of EGFR-TKI resistant patients during the treatment. The T790M mutation, on the other hand, occurred in a patient who had previously been untreated with EGFR TKIs, implicating an additional role in intrinsic resistance to first- and second-generation TKIs [25,26,27].

### 2.2. Third-Generation EGFR-TKIs

Because of the high prevalence of T790M mutation and the low efficacy of first- and second-generation EGFR inhibitors due to the steric hindrance effected by T790M [28], third-generation EGFR-TKI was initially developed. The FDA has authorized osimertinib, a third-generation irreversible EGFR-TKI, to treat patients with the EGFR T790M mutation who have developed resistance to first- and second-generation EGFR TKIs [29]. Recently, the FLAURA trial demonstrated the superiority of osimertinib over gefitinib or erlotinib in the first-line setting of EGFR-mutant NSCLC as shown in an improved mOS and mPFS for advanced EGFR mutant NSCLC [30,31]. Because of the limited penetration of first- and second-generation EGFR TKIs into the blood–brain barrier, about 40% of NSCLC patients with EGFR mutations develop CNS metastases. Notably, in the phase I trial (BLOOM), osimertinib revealed considerable treatment benefits in the CNS and a tolerable safety profile in patients with leptomeningeal metastases from EGFR-mutated advanced NSCLC [32]. The evidence showed no T790M mutation after applying osimertinib as the first-line setting [33]. Thus, T790M mutation as a resistance mechanism has become less clinically significant. Instead, acquired resistance in other EGFR-dependent and EGFR-independent bypass pathways was developed. Despite the fact that the cysteine-797 (C797) residue in the ATP binding site of the EGFR kinase is the target of third-generation EGFR TKIs, in preclinical models and clinical samples, acquired EGFR T790M/C797S mutation was eventually developed [34] and C767 mutations have been observed in 15% of second-line osimertinib patients and 7% of first-line osimertinib patients [35].

As a result, there is an urgent need to identify inhibitors that can bind to sites other than the ATP binding cleft of the EGFR-TK-domain in order to overcome the resistance related to third-generation EGFR inhibitors. To date, a novel class of allosteric mutant-selective fourth-generation EGFR-TKIs which can bind to the site other than the ATP binding cleft of EGFR, have been discovered to overcome third-generation EGFR-TKIs resistance and introduced for clinical evaluation [36].

## 3. Proteomics in the Study of Molecular Mechanisms of Acquired Resistance of EGFR-TKI in NSCLC

Long-term exposure to inhibitors exerts selective pressure on tumor cells to become resistant to therapy, including EGFR-TKIs [37]. Proteomic analysis has grown in importance in molecular sciences since it provides large-scale protein information, including protein expression profiles in drug-resistant cancer cells. Mass spectrometry-based proteomics has evolved into a promising technique for investigating numerous pathways of drug resistance in cancer cells, allowing for the global identification and quantification of proteins related to drug resistance [17,38]. Such approaches are being employed to uncover the fundamental differences between sensitive and resistant cancers, which reveal drug resistance mechanisms and biomarkers for predicting response to the regimen [17,38], leading to the development of novel therapeutics that target proteins that are specifically expressed in resistant cancers [39].

In recent decades, proteomic studies of EGFR-TKI-resistant NSCLC have been largely investigated (Table 2). In the field of proteomics, MS-based protein analysis technology has dominated. In most studies, this approach, combined with either labeling or non-labeling quantitation methods, was extensively used to reveal global changes in the proteome and phosphoproteome following EGFR-TKI treatment in NSCLC. Due to the limitation of surgically resected tissues from patients with EGFR-TKI resistance, the proteome and phosphoproteome profiles of EGFR-TKI resistance were widely investigated in the in vitro model. The majority of the research compared EGFR-TKI-resistant cell lines to EGFR-TKI-sensitive cell lines. 

According to the findings of proteomic and phosphoproteomic investigations of acquired EGFR-TKI resistance, a wide range of dysregulated proteins and signaling pathways are involved in the activation of bypass and crosstalk signaling pathways and changes in histological phenotype from NSCLC to SCLC or epithelial to mesenchymal transition (EMT). Furthermore, the study of proteome profiles of liquid biopsy samples unveiled the potential biomarkers for NSCLC patient stratification that are currently being applied in the clinic.

### 3.1. Bypass and Downstream Pathway Activation

#### 3.1.1. Aberrant Expression of RTK

The investigation of the tyrosine phospho-proteome of EGFR-TKI-sensitive PC9 cells vs. erlotinib-resistant PC9GR cells showed activation of several receptor tyrosine kinases (RTKs), including Met, IGF, and AXL signaling pathways, as shown in Figure 1 [58]. Even though amplification of the *MET* gene is reported in tumors resistant to first-line erlotinib, gefitinib, or afatinib and osimertinib [60], the activation of Met signaling observed in this phospho-proteomic study is independent of *MET* gene amplification. The study also unveiled the extensive signaling crosstalk involving the acquired resistance mechanism to the EGFR-TKI treatment. Almost half of the statistically significant phospho-tyrosine peptides were increased in response to the treatment of erlotinib.

#### 3.1.2. SRC Signaling Pathway

The acquired resistance mechanism of NSCLC was explored through the use of a TK activity-representing peptide library-based multiple reaction monitoring (TARPL-MRM) to determine tyrosine kinase (TK) activity in response to osimertinib treatment [41]. The results indicated a rewiring of TK activity and that the phosphorylated activation loops of SRC family proteins, including SRC, ACK, FER, and FYN, were significantly increased in H1975 cells treated with osimertinib at different time points. Network analysis of TK alteration of sensitive and resistant cells also confirmed the SRC family was a key mediator in the resistant NSCLC cells.

#### 3.1.3. FGFR-Akt

The proteome analysis revealed an increased expression of the receptor tyrosine kinase AXL in erlotinib-resistant cell lines and aberrant expression of FGFR1, FRS-2, and PRAS40, indicating the activation of the FGFR1-Akt pathway [44]. A combination of FGFR1 and Akt inhibitors synergistically inhibited EGFR-TKI-resistant NSCLC cells with FGFR1 overexpression. The tumor growth rate was significantly inhibited upon the co-treatment of an FGFR1 inhibitor and an Akt inhibitor in EGFR-TKI-resistant NSCLC xenograft models. Furthermore, high FGFR1 mRNA expression levels were a statistically significant prognostic marker for progression-free survival of EGFR-TKI-treated patients.

#### 3.1.4. FAK Signaling

According to array-based and MS-based proteomic analysis, FAK signaling has been reported for its involvement in the acquired EGFR-TKI resistance mechanism of the first- and second-generation of EGFR-TKI. The proteome profiler array was used to examine the expression of human soluble receptors and related proteins in gefitinib-sensitive parent cells and gefitinib-resistant cell lines [48]. The study demonstrated that osteopontin (OPN) was the most significantly overexpressed in gefitinib-resistant NSCLC cells. OPNs have been shown to interact with various integrins through RGD-mediated integrin recognition sequences. OPN contributes to the acquired EGFR-TKI resistance mechanism by increasing integrins αv and β3 levels. The downstream FAK/AKT and ERK signaling pathways were subsequently activated, which enhanced NSCLC cell proliferation [48]. A p-FAK inhibitor dramatically improved the susceptibility of gefitinib-resistant cells to gefitinib. A combination of gefitinib and p-FAK inhibitors effectively inhibited gefitinib-resistant cell growth. The findings were further verified in a mouse xenograft model, where tumor growth was suppressed by a combination regimen more potent than a single gefitinib treatment.

In addition, the MS-based quantitative proteomic analysis of osimertinib-sensitive parent cells and osimertinib-resistant cell lines demonstrated that LAMA5 (Laminin α5) was the highest fold change in osimertinib-resistant cells [42]. Enhanced expression of LAMA5 was associated with high plasma IL-6 levels and in osimertinib-resistant NSCLC cells with high IL-6 levels. Furthermore, the activation of FAK, a downstream effector of LAMA5, was observed exclusively in osimertinib-resistant cells with high IL-6 levels. A combination treatment of osimertinib and ibrutinib efficiently reversed the drug resistance in osimertinib-resistant cell lines, through inhibition of IL-6 and lamininα5/FAK signaling [42].

#### 3.1.5. PI3K/Akt/mTOR and MAPK/Erk Signaling

Integrative analysis of the proteome and phosphoproteome has been used to study aberrations of signaling pathways in both second- and third-generation EGFR-TKI. Mulder and colleagues employed a multi-omic approach to investigate the alteration of the proteome, kinome, and phosphoproteome profiles of NSCLC cells during afatinib treatment (1 to 7 days) [52]. Upon the initial afatinib treatment, NSCLC cells adapt to afatinib inhibition by using Ca^2+^/calmodulin-related signaling and adhesion signaling pathways as a resistance mechanism. Phosphoproteomic data also demonstrated reactivation of the PI3K/mTOR and MEK/ERK signaling pathways within days after afatinib treatment. A combination of mTORC1 inhibitor (rapamycin) and afatinib had cytostatic effects on the growth of NSCLC cells, in which cell growth was significantly inhibited with no alteration in the amount of apoptotic cells. Interestingly, the effects of MEK inhibitor (selumetinib) and afatinib co-treatment induced apoptosis in NSCLC cells, which is beneficial for use as an anti-tumor agent. 

The global proteome and phosphoproteome of the third generation of EGFR-TKI resistance mechanism of NSCLC were investigated in osimertinib- and rociletinib-resistant cells vs. sensitive cells [46]. The phosphorylation levels of phosphatase PTPN11 (SHP2) important sites were reduced in all resistant cell lines, resulting in inactive phosphatase activity and consequent activation of PI3K/AKT pathways and suppressed RAS/MAPK signaling [46]. The treatment of dactolisib, a dual PI3K/AKT and mTOR inhibitor, as a combination agent with osimertinib inhibited resistant NSCLC cells both in vitro and in animal models. Persistent activation of ERK signaling was also observed in osimertinib-resistant cell lines after osimertinib treatment [54]. The study performed a phospho-kinase array to analyze 43 different kinase phosphorylation patterns in both parental and osimertinib-resistant NSCLC cells in the presence of osimertinib. The phosphorylation of WNK1, a regulator of MAPK in EGFR signaling and involved in cell proliferation, was induced after osimertinib treatment in osimertinib-resistant cells [54]. A combination of MEK inhibitor and osimertinib efficiently inhibited resistant cell viability and induced apoptosis by suppressing ERK phosphorylation. Furthermore, the combination of two inhibitors significantly inhibited tumor growth more potently than the treatment with a single agent.

### 3.2. Histological Transformation

Proteomic and phosphoproteomic studies of the acquired resistance mechanisms of NSCLC cells to multiple generations of EGFR-TKI revealed a plasticity transformation from epithelial to mesenchymal cells in EGFR-TKI-resistant NSCLC cells.

The proteome and phosphoproteome study of the first generation of EGFR-TKI has been performed using reverse phase protein arrays (RPPA), and immunoaffinity enrichment of pTry phosphopeptides combined with LC-MS/MS. Integrated proteomic analysis of RPPA, gene expression, and drug resistance analysis demonstrated that mesenchymal cancer cells were more resistant to EGFR-TKI and PI3K/Akt pathway inhibitors, independent of EGFR mutation status, compared with epithelial cells [59]. The overexpression of the Axl protein was found in mesenchymal cell types compared with epithelial cells [59]. Proteomic analysis of T790M-negative erlotinib-resistant NSCLC cell lines revealed a link between EMT and the EGFR-TKI resistance mechanism, with mesenchymal markers AXL and ZEB1 overexpression and epithelial markers E-cadherin and -catenin expression lower in EGFR-TKI resistant cells [49]. The tyrosine phospho-proteome analysis also demonstrated that multiple Src/FAK pathway kinases were aberrantly phosphorylated in mesenchymal cells [57]. Using unbiased drug sensitivity screening, the Abl/Src inhibitor dasatinib was demonstrated as the most potent anti-cancer agent for erlotinib-resistant mesenchymal cells. Furthermore, using an integrative analysis of transcriptomic, proteomic, and drug screening data, the activation of the yes-associated protein (YAP) and forkhead box protein M1 (FOXM1) axis has been found as a driver of EMT-associated EGFR TKI resistance and upregulated the expression of spindle assembly checkpoint (SAC) proteins [49].

Quantitative proteomic and phosphoproteomic analyses of the third generation of EGFR-TKI were performed in resistant EGFR-mutant NSCLC cells and sensitive cells. iTRAQ-based quantitative proteomics and whole-transcriptome sequencing demonstrated that NSCLC cells harboring the EGFR C797S mutation are associated with a mesenchymal-like cell state with elevated expression of AXL receptor tyrosine kinase [47]. Enrichment analysis of the biological processes of differentially expressed proteins indicated a strong relationship with EMT, cytoskeletal rearrangement, and migratory and invasive properties. The inhibition of AXL effectively suppressed the growth of NSCLC cells with the EGFR C797S [47]. Global SILAC quantitative mass spectrometry demonstrated that expression levels of several translational regulator proteins, including EIF proteins and EMT signature proteins, were much more altered in rociletinib-resistant cells than in osimertinib-resistant cells [46]. CDH1 expression, which is linked with the epithelial state, was reduced in resistant cells.

### 3.3. Miscellaneous Pathway

#### 3.3.1. Autophagy

Autophagy is an intracellular catabolic process that eliminates cytoplasmic materials or malfunctional components via a lysosome-dependent mechanism. Translation-related proteins have been reported for their roles in erlotinib resistance. The eIF3c, a eukaryotic translation initiation factor (eIF), was highly upregulated in T790M-negative PC9/ER and mechanistically enhanced autophagic activity through increasing an autophagy marker, LC3B-II [50]. Erlotinib-induced autophagy is inhibited by eIF3c suppression, showing that eIF3c is a critical regulator of erlotinib-induced autophagy [50]. Similarly, applying the quantitative global proteome and diGly proteomics, which combine antibody-based capture of “diGly remnant” peptides and SILAC, the results showed that thousands of differentially expressed proteins and ubiquitylation were associated with gefitinib resistance [53]. Furthermore, HMGA2 and ALOX5, which are involved in promoting tumor metastasis [61] and aberrantly expressed in several tumor types [62], respectively, were chosen and subsequently validated. HMGA2 overexpression or ALOX5 knockdown suppressed gefitinib resistance in NSCLC cells by inhibiting autophagy [53].

#### 3.3.2. Antigen Presenting Pathway

Employing pan-HLA class I antibody-based affinity purification-mass spectrometry (AP-MS) provided the evidence that osimertinib resistance in EGFR mutant lung cancer resulted in widespread suppression of HLA peptide processing and presentation, which was demonstrated by a decrease in the HLA class I-presented immunopeptidome as well as the antigen presentation core complex (e.g., TAP1 and ERAP1/2) [43]. Furthermore, through integrated pathway analysis, the alteration of the immunoproteasome, several key elements in autophagy, caspases, or phagosome signaling affected the source of antigen in osimertinib-resistant lung adenocarcinoma cells [43].

#### 3.3.3. Metabolism

iTRAQ-based quantitative proteomic analysis has been used for the identification of differentially expressed proteins among gefitinib-resistant PC9/GR cells and the corresponding parental PC9 cells. Nicotinamide N-methyltransferase (NNMT) was the most significantly upregulated in PC9/GR cells [40]. The upregulation of NNMT in NSCLC patients who received EGFR-TKI treatment was associated with lower progression-free survival and poor survival outcomes. the knockdown of NNMT in PC9/GR and HCC827/GR cells significantly increased sensitivity to gefitinib and erlotinib, and induced cell apoptosis. Furthermore, the overexpression of NNMT significantly decreased the sensitivity of NSCLC cells to the third-generation EGFR TKI inhibitor, osimertinib. The upregulation of NNMT mediates EGFR-TKI resistance by regulating the glycolysis mechanism of NSCLC cells by increasing c-myc expression via SIRT1-mediated c-myc deacetylation. In NSCLC cells, the combination of NNMT inhibitor and EGFR-TKI effectively overcomes EGFR-TKI resistance [40].

The MS-based proteomic study of the third-generation EGFR-TKI, almonertinib, identified increased expression of glutamine transporter (SLC1A5) in NSCLC cells treated with almonertinib (57). The inhibition of glutamine influx by siRNA knockdown of SLC1A5 and SLC1A5 effectively decreased NSCLC cell proliferation and glutamine uptake. A combination of the SLC1A5 inhibitor and almonertinib could improve in vivo anti-tumor activity with no severe liver or renal toxicity.

#### 3.3.4. Post-Translational Modification of EGFR

Understanding the functional significance of glycosylation-mediated disease necessitates extensive characterization of the glycoproteome, which is extremely difficult due to the intrinsic complexity of glycoproteins. Recent studies have found that glycosylation has a significant role in lung cancer resistance. For example, sialylation of EGFR has been demonstrated to alter susceptibility to TKIs [63,64]. However, a comprehensive glycoproteome study in a lung cancer cell is yet substantially unexplored. Waniwan et al. implemented lectin nanoprobe-based affinity mass spectrometry for complementary glycotope-specific enrichment and site-specific glycosylation analysis of the glycoproteome [51]. They discovered a significant quantity of glycopeptides, particularly fucosylated glycopeptides, in the resistant PC9-IR cells [51]. Aberrant fucosylation mediates EGF-mediated cellular growth response and gefitinib sensitivity by influencing either the binding affinity of EGFR to the EGF ligand or the ability of the EGFR to dimerize [51].

## 4. Proteomics in Serum Biomarker Discovery of EGFR-TKI Resistance in NSCLC

Liquid biopsy proteome profiling has been intensively studied for the discovery of biomarkers for predicting and monitoring EGFR-TKI response in NSCLC patients. The proteomic approach, which included a gel-based [65], antibody-based [66], and MS-based technique [67], was performed to analyze biofluid samples from NSCLC patients who responded differently to EGFR-TKI as summarized in (Table 3).

The use of 2D-DIGE for studying serum proteome profiles at the baseline and progression of the disease showed that alpha-1-antitrypsin (AAT) was highly upregulated in progressive diseases (PD) compared to baseline levels in advanced NSCLC patients treated with erlotinib or gefitinib [65]. AAT1 levels were lower in patients with partial responses to EGFR-TKI.

The proteome profiles of the serum of advanced NSCLC patients treated with erlotinib were determined using a 41,472 antibody microarray and LC-MS/MS [66]. The results showed an association between levels of isoform 2 of fibrinogen alpha chain (FGA2) and EGFR-TKI response. FGA2 levels were decreased in the PR group but increased in the PD group. Interestingly, FGA2 was not detected in lung cancer cells but in hepatocytes. Hepatocellular carcinoma cells treated with erlotinib decreased the expression and secretion levels of FGA2. This finding might at least in part explain the fluctuations of serum FGA2 levels upon the treatment of erlotinib in NSCLC patients.

Proteome profiles of pleural effusion (PE) were determined using iTRAQ labeling coupled with LC/MS-MS [67]. The specimens were derived from NSCLC patients carrying EGFR mutations with a differential response to EGFR-TKI treatment. PE levels of soluble cadherin-3 (sCDH3) were higher in patients resistant to EGFR-TKI. Serum levels of sCDH3 were also determined at baseline and 1 month after EGFR-TKI treatment. The results demonstrated lower sCDH3 levels in PR patients. Furthermore, serum sCDH3 showed its independent prognostic power, in which sCDH3 levels were linked with the progression-free survival (PFS) of NSCLC patients.

The most extensive studies on the discovery of EGFR-TKI response predictive markers are the detection of unique proteomic spectra using high throughput MALDI-TOF mass spectrometry. The use of the MALDI MS algorithm based on eight distinct m/z features called “VeriStrat” was applied primarily in the baseline serum of NSCLC patients for stratification of NSCLC patients into good vs. poor responders for the treatment of Erlotinib or Gefitinib [68]. A later study demonstrated the efficiency of VeriStrat in monitoring gefitinib responses, in which “good” VeriStrat classification was linked to longer overall survival independently of other clinical factor confounders [69]. Specific proteins were further identified from the eight MALDI TOF MS signals between poor and good responders using LC MS/MS [70]. Serum amyloid A protein 1 (SAA1) was higher expressed in the plasma of the NSCLC patients who poorly responded to gefitinib treatment, in which case SAA1 generated four out of the eight MS mass signals composing the VeriStrat algorithm. This test has been commercially launched as VeriStrat, and its clinical relevance has been validated in clinical trials [71,72,73,74,75,76].

Additionally, Yang et al. applied MALDI-TOF-MS and ClinProTools software to identify serum peptides and proteins associated with EGFR gene mutation status in stage IIIB or IV NSCLC patients with EGFR gene TKI-sensitive mutations and wild-type EGFR genes [77]. The serum proteomic classifier established was examined for EGFR gene mutation status and verified in an independent validation cohort, demonstrating high concordance and sensitivity with tumor biopsies. Furthermore, the classifier was also consistent with tests in tumor tissue for identification of response to EGFR-TKI treatment [77].

**Table 3 ijms-24-04827-t003:** Summary of proteomic studies in EGFR-TKI resistance in biofluid specimens of NSCLC.

Study	Samples	Discovery Cohort/Training Set	Sampling Time	EGFR-TKI	Protein Biomarker/ Biomarker	Sample Pre-Processing	Proteomic Technique	Finding
Hsiao 2020 [67]	Pleural effusion	Advanced lung adenocarcinoma with EGFR mutation with differential response to EGFR-TKI (N = 23) and patients with tuberculosis (N = 10)	-	Gefitinib, Erlotinib, Afatinib	Cadherin-3 (CDH3)	Multiple Affinity Removal System (MARS) Affinity Column	iTRAQ/2D LC MS-MS	[EGFR-TKI response predictive marker, Prognostic marker] The PE level of soluble CDH3 (sCDH3) was increased in patients with resistance. The altered sCDH3 serum level reflected the efficacy of EGFR-TKI after 1 month of treatment (*n* = 43). Baseline sCDH3 was significantly associated with PFS and OS in patients with ADC after EGFR-TKI therapy (*n* = 76). Moreover, sCDH3 was positively associated with tumor stage in non–small cell lung cancer (*n* = 272).
Shang 2019 [66]	Serum	Advanced lung adenocarcinoma with EGFR mutation who had partial response after 2 cycles of first-line erlotinib (N = 9) and heathy control (N = 9)	Baseline, PR, and PD)	Erlotinib	Isoform 2 of fibrinogen alpha chain (FGA2)	-	Antibody microarray/immunoprecipitation/ LC-MS/MS (Q-Orbitrap)	[EGFR-TKI response predictive marker] serum FGA2 level was correlated with EGFR-TKI response (*p* < 0.05).
Zhao 2013 [65]	Serum	Advanced lung adenocarcinoma with long PFS (N = 18)	Baseline and PD	Erlotinib, Gefitinib	a1-antitrypsin (AAT)	Liquid chromatographic column	2D-DIGE/MALDI-TOF/TOF	[EGFR-TKI response predictive marker] AAT was upregulated in PD compared with baseline, with an average ratio of 1.68 (P¼0.0017), and Western blot analysis showed that AAT was downregulated in PR.
Milan 2012 [70]	Serum	Advanced NSCLC patients	Baseline	Gefitinib	Serum amyloid A protein 1 (SAA1)	Agilent Multiple Affinity Removal System	2DE/MALDI-TOF/LC-MS/MS (Q-TOF)	[EGFR-TKI response predictive marker]
Buttigliero 2019 [76]	Serum	Advanced NSCLC treated in the second or third line with tivantinib plus erlotinib (T+E) compared with placebo plus erlotinib (P+E)	Baseline	Erlotinib	Proteomic spectra		VeriStrat	[EGFR-TKI response predictive marker, Prognostic marker] Phase III clinical trial
Wu 2013 [78]	Serum	Advanced NSCLC patients (N = 24)	Baseline	Erlotinib, Gefitinib	Serum proteomic classifier	MB-WCX kits	MALDI-TOF	[EGFR-TKI response predictive marker]
Lazzari 2012 [69]	Plasma	NSCLC Patients (N = 111)	baseline, after 1 month and concomitantly with CT scan evaluation performed every other month until withdrawal from treatment with EGFR TKIs for either toxicity or progression.	Gefitinib	Proteomic spectra	-	MALDI-TOF (Veristrat)	[EGFR-TKI response predictive marker]
Taguchi 2007 [68]	Serum	NSCLC patients (N = 139)	Baseline	Erlotinib, Gefitinib	Proteomic spectra	-	MALDI-TOF	[EGFR-TKI response predictive marker] This MALDI MS algorithm was not merely prognostic but could classify NSCLC patients for good or poor outcomes after treatment with EGFR TKIs. This algorithm may thus assist in the pretreatment selection of appropriate subgroups of NSCLC patients for treatment with EGFR TKIs.

Abbreviation: PR: partial response, PD: disease progression/progressive disease, PFS: progression-free survival.

## 5. Clinical Applications

Given the rapid progression of disease in patients with acquired resistance to EGFR-TKI treatment, there is a significant unmet need for novel therapeutic alternatives. Based on protein and pathway alteration from the proteome and phosphoproteome analysis, we summarized targeted proteins and potential drugs that have been tested in clinical studies.

### 5.1. AXL

AXL is overexpressed in many types of cancer, including NSCLC, breast, gastric, colorectal, and prostate cancer [79]. The overexpression of AXL has been linked to drug resistance to a variety of inhibitors, including an EGFR inhibitor [80]. In NSCLC, AXL upregulation has been associated with EMT, with AXL being overexpressed in mesenchymal cancer cells compared to epithelial cancer cells [40,43,45,52,54,55,59]. The downregulation of AXL expression inhibited EMT while increasing the response to EGFR-TKI [81]. Several studies on the efficacy of agents targeting AXL, including small molecule inhibitors, monoclonal antibodies, and antibody-drug conjugates, have been conducted in preclinical and clinical phases [82].

In preclinical studies both in vitro and in vivo, the expression of AXL was induced in response to the treatment of EGFR-TKIs in NSCLC cells carrying an EGFR mutation. The combination of AXL inhibitors and EGFR-TKIs could synergistically overcome this resistance [83,84,85]. Several potent AXL inhibitors have recently been evaluated in early-phase clinical trials [82]. A phase I clinical study of the combination of DS-1205c with Gefitinib for metastatic or unresectable EGFR-mutant NSCLC (NCT03599518) demonstrated no serious adverse events directly related to DS-1205c [86]. A phase I/II trial of the oral selective AXL inhibitor bemcentinib (BGB324) in conjunction with erlotinib in patients with advanced EGFR mutation NSCLC (NCT02424617) revealed the feasibility and well-acceptability of this combination, with benefit found in a subgroup of patients who had progressed on an EGFR inhibitor or were receiving erlotinib simultaneously in remission in the first line [87]. However, a clinical trial combining ASP2215 and erlotinib in EGFR-positive NSCLC patients following EGFR inhibitor treatment (NCT02495233) was halted due to significant serious adverse effects associated with the combination medication.

### 5.2. SRC

SRC is a member of the non-receptor tyrosine kinase family (Src Family Kinases, SFKs) and plays a critical role in cell adhesion, invasion, proliferation, survival, and angiogenesis during tumor development [88]. Src activation is important in the acquisition and maintenance of resistance to EGFR inhibitors in lung cancer [41], in addition to the well-established involvement of Src kinases in tumor growth. Interestingly, EGFR-TKI-resistant cells had drastically reduced cell survival and migration after treatment with the SFK inhibitor dasatinib, demonstrating that Src inhibitors may overcome EGFR inhibitor resistance in lung cancer cells [89].

In this phase I trial (NCT0199998), Dasatinib revealed high tolerance in cancer patients who progressed after EGFR inhibitors and feasibility in advanced NSCLC at biologically active dosages in conjunction with afatinib. Despite this, pleural effusion remained a significant major adverse effect. In an open-label, dose-escalation phase I/II trial (NCT01999985) with two-stage expansion, This combination demonstrated a low toxicity profile and lowered the incidence of EGFR mutations and T790M variant alleles in cell-free DNA, but no objective clinical responses were reported [90]. Another clinical trial (NCT02954523, phase I/II) evaluated the effects of the third-generation EGFR-TKI, osimertinib in combination with dasatinib, in EGFR mutant NSCLC patients who developed resistance to the first-generation EGFR-TKIs and assessed serum biomarkers to monitor clinical outcomes upon Src inhibitor treatments. Although the combination of osimertinib and dasatinib had anti-tumor efficacy in patients with EGFR-mutant NSCLC in the front-line setting, the treatment was limited by chronic toxicities, which were primarily due to dasatinib [91].

### 5.3. PI3K Signaling Pathway

The alterations of the PI3K-AKT-mTOR pathway occur through the activation of tyrosine kinase receptors, PIK3CA amplification, and mutations in downstream signaling [92]. Clinical studies have shown that EGFR mutant patients with PI3K pathway activation have a shorter PFS and a lower OS [93,94,95]. The mTOR inhibitors, including everolimus, have been approved for cancer treatment including neuroendocrine tumors and, as a combination therapy, HER2-positive breast cancer, as well as certain tuberous sclerosis complex-related tumors [96].

In preclinical studies, everolimus was shown to overcome EGFR drug resistance and provide a cooperative impact with EGFR inhibitors in various human cancer cell lines resistant to EGFR inhibitors [97]. Everolimus synergized with gefitinib to restore the EGFR-TKI resistance in NSCLC cell lines [98,99]. The combination therapy of everolimus and EGFR-TKIs was evaluated for feasible dosages for tolerable toxicity and disease control in NSCLC patients carrying EGFR mutations [100,101]. Everolimus was used as a second-line treatment in a patient with an EGFR mutation who had failed to respond to EGFR-TKI and had tumor regression [102].

## 6. Conclusions

Acquired resistance remains a challenge for the effective treatment of NSCLC. Due to the limited availability of repeated tissue biopsies, the proteomics and phosphoproteomics analysis of cancer cells with in vitro treatment of EGFR-TKI has been widely performed for the discovery of therapeutic targets and biomarkers. The validation of the proteome and phosphoproteome data in in vitro and in vivo models is crucial for identifying the best candidates for further clinical studies. Furthermore, an integrative analysis of genomics and proteomics will provide more insight into the system biology of EGFR-TKI resistance mechanisms in NSCLC patients.

## Figures and Tables

**Figure 1 ijms-24-04827-f001:**
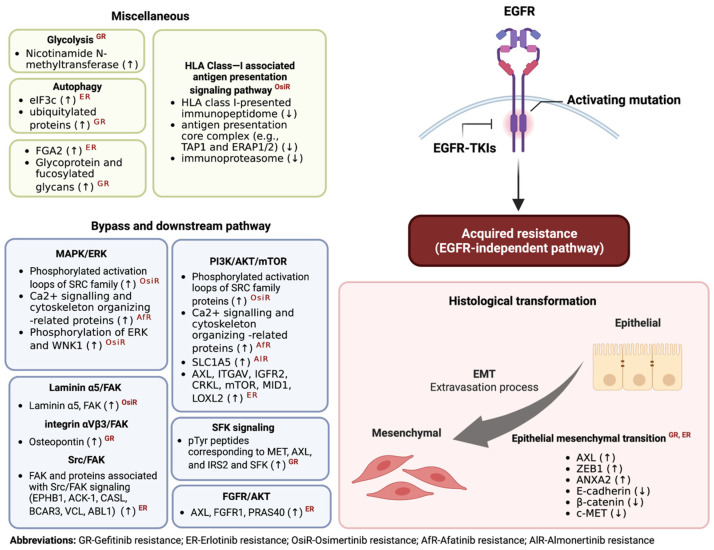
Summary of dysregulated proteins in proteomics studies of acquired resistance of EGFR-TKI through EGFR-independent pathway in NSCLC. Up arrows indicate upregulated proteins. Down arrows indicate downregulated proteins.

**Table 1 ijms-24-04827-t001:** Clinically used EGFR-TKIs in NSCLC.

Generation	Drug	Chemical Classification	Targets	Clinical Application for NSCLC	Approved Years
First generation EGFR-TKIS	Gefitinib	quinazolinamine	EGFR	Unselected NSCLC First-line therapy: Metastatic EGFR-sensitizing mutant	2002 2015
	Erlotinib	quinazolinamine	EGFR, EGFR (del19), EGFR (L858R)	Locally advanced or metastatic NSCLC First-line therapy: Advanced EGFR-sensitizing mutant	2004 2013
	Icotinib	quinazolinamine	EGFR (L858R)	Locally advanced or metastatic NSCLC First-line therapy: Metastatic EGFR-sensitizing mutant	2011 2014
Second generation EGFR-TKIS	Afatinib	quinazolinamine	EGFR, EGFR (L858R/T790M), HER2, HER4	Metastatic EGFR-sensitizing mutant	2013
	Dacomitinib	quinazolinamine	EGFR, EGFR (del19), EGFR (L858R), HER2, HER4	First-line therapy: Metastatic EGFR-sensitizing mutant	2018
Third generation EGFR-TKIS	Osimertinib	aminopyrimidines	EGFR, EGFR (del19), EGFR (L858R), EGFR (T790M)	EGFR-T790M mutation First-line therapy: Metastatic EGFR-sensitizing mutant Adjuvant therapy	2015 2018 2020
	Almonertinib	aminopyrimidines	EGFR (del19), EGFR (L858R), EGFR (T790M)	EGFR-T790M mutation	2020
	Furmonertinib	aminopyrimidines	EGFR (del19), EGFR (L858R), EGFR (T790M)	Locally advanced or metastatic NSCLC with EGFR T790M mutation	2021

**Table 2 ijms-24-04827-t002:** Summary of proteomic studies in EGFR-TKI resistant in NSCLC.

Study	Dysregulated Proteins	Altered Pathways	Mechanisms	EGFR-TKIs (Generation)	Samples	Proteomic and Quantitation Techniques
Wang 2022 [40]	Nicotinamide N-methyltransferase (↑)	Glycolysis (↑)	Miscellaneous	Gefitinib (1st)	PC9 vs. PC9/GR	LC-MS/MS, iTRAQ-labeling
Hou 2022 [41]	phosphorylated activation loops of SRC family proteins, such as SRC, ACK, FER, and FYN (↑)	The MAPK/ERK pathway, PI3K/AKT signaling (↑)	Bypass and downstream pathway activation	Osimertinib (3rd)	H1975 vs. H1975OsiR	Phosphopeptide Enrichment LC-MS/MS
Li 2022 [42]	Laminin α5, FAK (↑)	PI3K-AKT, Laminin α5/FAK signaling (↑)	Bypass and downstream pathway activation	Osimertinib (3rd)	PC9/GR vs. PC9/GROsiR	LC-MS/MS
Qi 2021 [43]	HLA class I-presented immunopeptidome, antigen presentation core complex (e.g., TAP1 and ERAP1/2), and immunoproteasome (↓)	immunoproteasome and autophagy cascades (↓)	Miscellaneous	Osimertinib (3rd)	PC9 vs. PC9/OsiR H1975 vs. H1975/OsiR	LC-MS/MS, SILAC labeling
Terp 2021 [44]	The receptor tyrosine kinase AXL, FGFR1, PRAS40 (↑)	FGFR1-Akt pathway (↑)	Bypass and downstream pathway activation	Erlotinib (1st)	HCC827 vs. HCC827/ER	LC-MS/MS, iTRAQ-labeling
Liu 2021 [45]	SLC1A5 (↑)	PI3K-AKT (↑)	Bypass and downstream pathway activation	Almonertinib (3rd)	H1975 cell treated with Almonertinib vs. blank control	LC-MS
Zang 2021 [46]	Too many (↑)	PI3K/AKT pathways, EMT (↑)	Bypass, downstream pathway activation, and histological transformation	Osimertinib or Rociletinib (3rd)	H1975 vs. H1975/OsiR H1975 vs. H1975/COR	LC-MS/MS, SILAC labeling
Wang 2020 [47]	AXL (↑)	EMT, cytoskeletal reorganization, and migratory and invasive properties. (↑)	Histological transformation	3rd Generation	H1975 vs. H1975-MS35	LC-MS/MS, iTRAQ-labeling
Fu 2020 [48]	Osteopontin (↑)	integrin αVβ3/FAK signaling pathway (↑)	Bypass and downstream pathway activation	Gefitinib (1st)	PC9 vs. PC9/GR HCC827 vs. HCC827/GR	Proteome profiler array
Nilsson 2020 [49]	AXL and ZEB1 (↑) E-cadherin and 𝛽-catenin, c-MET (↓)	EMT, cytoskeletal reorganization, and migratory and invasive properties. (↑)	Histological transformation	Erlotinib (1st)	HCC827 vs. HCC827/ER HCC4006 vs. HCC4006/ER	Reverse phase protein array
Shintani 2018 [50]	eukaryotic translation initiation factor 3 subunit C (eIF3c) (↑)	Autophagy (↑)	Miscellaneous	Erlotinib (1st)	PC9 vs. PC9/ER	LC-MS/MS
Waniwan 2018 [51]	Glycoprotein and fucosylated glycans (↑)	-	Miscellaneous	Gefitinib (1st)	PC9 vs. PC9-IR	lectin−magnetic nanoprobe/LC-MS/MS
Mulder 2018 [52]	Ca2+ signaling and cytoskeleton organizing -related proteins (↑)	mTOR and MAPK signaling pathway (↑)	Bypass and downstream pathway activation	Afatinib (2nd)	PC9 treated with Afatinib	Phosphopeptide Enrichment LC-MS/MS
Li 2018 [53]	upregulated proteins or ubiquitylated proteins (↑)	Autophagy (↑)	Miscellaneous	Gefitinib (1st)	PC9 vs. PC9/GR	LC-MS/MS, SILAC labeling
Ku 2018 [54]	phosphorylation of ERK and WNK1 (↑) EGFR phosphorylation (↓)	ERK signaling (↑)	Bypass and downstream pathway activation	Osimertinib (3rd)	PC9 vs. PC9/OsiR	Proteome profiler array
Yi 2018 [55]	ANXA2 (↑)	EMT (↑)	Histological transformation	Gefitinib (1st)	HCC827 cultured with CAF vs. cultured with NF	2DE-MALDI-TOF/TOF MS
Jacobsen 2017 [56]	AXL, ITGAV, IGFR2, CRKL, mTOR, MID1, LOXL2 (↑)	PI3K-Akt-mTOR signaling pathway (↑)	Bypass and downstream pathway activation	Erlotinib (1st)	PC9 vs. PC9/ER	LC-MS/MS, SILAC labeling
Wilson 2014 [57]	FAK and proteins associated with Src/FAK signaling (EPHB1, ACK-1, CASL, BCAR3, VCL, ABL1) (↑)	Src/FAK pathway (↑)	Bypass and downstream pathway activation	Erlotinib (1st)	HCC827 parental vs. mesenchymal	Immunoaffinity enrichment of pTry phosphopeptides/LC-MS
Yoshida 2014 [58]	phosphopeptides corresponding to MET, AXL, and IRS2 and SFK (↑)	SFK signaling (↑)	Bypass and downstream pathway activation	Gefitinib (1st)	PC9 vs. PC9/GR	immunoaffinity purification of tyrosine-phosphorylated peptides LC/MS-MS
Byers 2013 [59]	AXL (↑) E-cadherin (↓)	EMT (↑)	Histological transformation	Erlotinib (1st)	NSCLC Mesenchymal cells vs. epithelial cells	RPPA

Abbreviations: PC9/GR—gefitinib-resistant PC9 cell; PC9/OsiR—osimertinib-resistant PC9 cell; PC9/GROsiR—gefitinib/osimertinib-resistant PC9 cell; PC9/ER—erlotinib-resistant PC9 cell; PC9-IR—TKI resistant PC9 cell; H1975/OsiR—Osimertinib-resistant H1975 cell; H1975/COR—Rociletinib-resistant H1975 cell; H1975-MS35-H1975 eGFR C797S knock-in clone; HCC827/GR—gefitinib-resistant HCC827 cell; HCC827/ER—erlotinib-resistant HCC827 cell; HCC4006/ER—erlotinib-resistant HCC4006 cell; PR—Partial response during EGFR-TKI treatment; PD—Progressive disease when patients acquired resistance; CAF—Cancer-associated fibroblast; NF—Normal fibroblast; vs.—versus; Up arrow—upregulated proteins or pathways; Down arrow—downregulated proteins or pathways.

## Data Availability

Not applicable.

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
