# Peer review of "The Role of Proteomics and Phosphoproteomics in the Discovery of Therapeutic Targets and Biomarkers in Acquired EGFR-TKI-Resistant Non-Small Cell Lung Cancer"

_ijms, 2023, doi:10.3390/ijms24054827_

Round 1
Reviewer 1 Report
The manuscript summarizes the recent advancement of proteomics approaches to identify new therapeutics targets for non-small cell lung cancer. Overall I find this paper comprehensive and well written. It provides a thorough overview of the EGFR-TKI therapeutics field, as well as proteomics study of EGFR-TKI resistance in NSCLC, which is valuable for the whole field. The organization of the paper is crystal clear and straightforward, making it very easy to understand.
Author Response
Thank you very much for your endorsement of our manuscript.
Reviewer 2 Report
Authors Sutpirat Moonmuang et al. present a review of the role of proteomics in the discovery of therapeutic targets and biomarkers in acquired EGFR-TKI-resistant non-small cell lung cancer. This work includes proteomic and phosphoproteomic analyses of non-small cell lung cancer (NSCLC) as well as proteomic analyses of biofluid samples associated with acquired resistance in response to different generations of EGFR-TKI. The authors provide an overview of the target proteins and potential drugs that have been tested in clinical trials and discuss the challenges in translating this discovery into future NSCLC treatment. The authors performed a detailed analysis of the literature data on this topic, including recent publications.
The manuscript is written consistently and logically and includes informative tables and figures. The summary of proteomic studies in EGFR-TKI resistant in NSCLC and in biofluid specimens of NSCLC, in particular, are quite nice, with rich detail that many readers should consider in their research design (Tables 2 and 3).
As a minor edit, I would recommend including these citations [1-4] to show how this review article differs from those previously published.
I would recommend this manuscript for publication in the International Journal of Molecular Sciences.
1. Wu, L.; Ke, L.; Zhang, Z.; Yu, J.; Meng, X., Development of EGFR TKIs and Options to Manage Resistance of Third-Generation EGFR TKI Osimertinib: Conventional Ways and Immune Checkpoint Inhibitors. Front Oncol 2020, 10, 602762.
2. Rotow, J.; Bivona, T. G., Understanding and targeting resistance mechanisms in NSCLC. Nat Rev Cancer 2017, 17, (11), 637-658.
3. Koulouris, A.; Tsagkaris, C.; Corriero, A. C.; Metro, G.; Mountzios, G., Resistance to TKIs in EGFR-Mutated Non-Small Cell Lung Cancer: From Mechanisms to New Therapeutic Strategies. Cancers (Basel) 2022, 14, (14).
4. Babuta, J.; Hall, Z.; Athersuch, T., Dysregulated Metabolism in EGFR-TKI Drug Resistant Non-Small-Cell Lung Cancer: A Systematic Review. Metabolites 2022, 12, (7).
Author Response
Comments from Reviewer 2
Comments and Suggestions for Authors
Authors Sutpirat Moonmuang et al. present a review of the role of proteomics in the discovery of therapeutic targets and biomarkers in acquired EGFR-TKI-resistant non-small cell lung cancer. This work includes proteomic and phosphoproteomic analyses of non-small cell lung cancer (NSCLC) as well as proteomic analyses of biofluid samples associated with acquired resistance in response to different generations of EGFR-TKI. The authors provide an overview of the target proteins and potential drugs that have been tested in clinical trials and discuss the challenges in translating this discovery into future NSCLC treatment. The authors performed a detailed analysis of the literature data on this topic, including recent publications.
The manuscript is written consistently and logically and includes informative tables and figures. The summary of proteomic studies in EGFR-TKI resistant in NSCLC and in biofluid specimens of NSCLC, in particular, are quite nice, with rich detail that many readers should consider in their research design (Tables 2 and 3).
As a minor edit, I would recommend including these citations [1-4] to show how this review article differs from those previously published.
I would recommend this manuscript for publication in the International Journal of Molecular Sciences.
- Wu, L.; Ke, L.; Zhang, Z.; Yu, J.; Meng, X., Development of EGFR TKIs and Options to Manage Resistance of Third-Generation EGFR TKI Osimertinib: Conventional Ways and Immune Checkpoint Inhibitors. Front Oncol 2020, 10, 602762.
- Rotow, J.; Bivona, T. G., Understanding and targeting resistance mechanisms in NSCLC. Nat Rev Cancer 2017, 17, (11), 637-658.
- Koulouris, A.; Tsagkaris, C.; Corriero, A. C.; Metro, G.; Mountzios, G., Resistance to TKIs in EGFR-Mutated Non-Small Cell Lung Cancer: From Mechanisms to New Therapeutic Strategies. Cancers (Basel) 2022, 14, (14).
- Babuta, J.; Hall, Z.; Athersuch, T., Dysregulated Metabolism in EGFR-TKI Drug Resistant Non-Small-Cell Lung Cancer: A Systematic Review. Metabolites 2022, 12, (7).
Response: Thank you very much for your suggestion. We have included the citations in the introduction section of the manuscript, as shown here, and marked up using “Track changes” in the main text.
Here, we emphasize the proteome and phosphoproteomic research of EGFR-TKI-resistant NSCLC cells, which gives significant details on the acquired resistance mechanisms for each EGFR-TKI generation. The mechanisms of EGFR-TKI resistance and prospective therapeutic approaches have been comprehensively described, including alterations in key signaling pathways as well as the metabolome and lipidome profiles of resistant cells [18-21]. In addition to the well-known acquired EGFR-TKI resistance, the proteome analysis sheds light on important resistant mechanisms, such as the posttranslational modifications of EFGR and antigen-presenting pathways. We also discuss the proteome analysis of biofluid samples, which have clinical potential as biomarkers for patient stratification and prognostic indication.
